# Progesterone-Induced Sperm Release from the Oviduct Sperm Reservoir

**DOI:** 10.3390/cells11101622

**Published:** 2022-05-12

**Authors:** Supipi Mirihagalle, Jennifer Rose Hughes, David Joel Miller

**Affiliations:** Department of Animal Sciences, Institute for Genomic Biology, University of Illinois, Urbana, IL 61801, USA; mirihag2@illinois.edu (S.M.); jrhughes@illinois.edu (J.R.H.)

**Keywords:** sperm, oviduct, glycans, sialic acid, Lewis X, CatSper, calcium, motility, oocyte, Fallopian tube

## Abstract

In mammalian females, after sperm are deposited in the reproductive tract, a fraction of sperm migrates to the lower oviduct (isthmus) and forms a sperm storage site known as the functional sperm reservoir. The interactions between sperm membrane proteins and oviduct epithelial cells facilitate sperm binding to the oviductal epithelium and retention in the reservoir. Sperm are bound by glycans that contain specific motifs present on isthmic epithelial cells. Capacitated sperm are released from the reservoir and travel further in the oviduct to the ampulla where fertilization occurs. For decades, researchers have been studying the molecules and mechanisms of sperm release from the oviductal sperm reservoir. However, it is still not clear if the release of sperm is triggered by changes in sperm, oviduct cells, oviduct fluid, or a combination of these. While there is a possibility that more than one of these events are involved in the release of sperm from the reservoir, one activator of sperm release has the largest accumulation of supporting evidence. This mechanism involves the steroid hormone, progesterone, as a signal that induces the release of sperm from the reservoir. This review gathers and synthesizes evidence for the role of progesterone in inducing sperm release from the oviduct functional sperm reservoir.

## 1. Introduction

After deposition in the female reproductive tract, mammalian sperm migrate to the lower oviductal region known as the isthmus and are retained by adhesion to the oviduct epithelial cells to form a functional sperm reservoir [1,2]. Because fertilization occurs in the upper oviduct, the ampulla, sperm must be released from the reservoir to move to the site of fertilization. The sperm reservoir serves several functions, including storage of sperm until ovulation [3], maintenance of sperm viability, delay of capacitation, and hyperactivated motility, Refs. [4,5,6] ultimately lengthening sperm lifespan [7,8] These functions also reduce polyspermy by limiting the number of sperm that reach the ovulated oocyte in the ampulla at any one time [9,10,11,12].

Long-term sperm storage in the female reproductive tract is observed in a wide variety of animals including insects, birds, amphibians, reptiles, and fish [13,14]. Although many animals store sperm in a reservoir, the location of the reservoir differs among species. Mammalian females possess an oviductal reservoir for sperm storage, insects possess spermatheca and seminal receptacles [15], birds possess sperm storage tubules [16], and amphibians possess spermatheca [17]. Female reptiles store sperm in the anterior vagina and infundibulum [18]. The main purpose of all of these sperm storage organs is to keep sperm viable until oocytes reach the site of fertilization so that reproduction is successful.

It is important to note that although the human female reproductive tract is capable of keeping sperm viable for about a week, a distinct pre-ovulatory sperm reservoir has not been observed in the Fallopian tube [19]. However, there is evidence showing human sperm associate with the endosalpinx under in vitro conditions and that oviduct cell secretions affect sperm motility [20]. Further, human sperm bind tightly to oviduct cells in vitro [21] and oviduct binding extends the sperm lifespan [22]. There is evidence of successful pregnancy after sperm deposition in the female reproductive tract five days before ovulation [23], longer than sperm survival in vitro. Therefore, it can be postulated that there exist sperm storage mechanisms in the human Fallopian tube [24].

In the mammalian sperm reservoir, oviduct epithelial cells contain glycans that facilitate the formation of the reservoir via binding of the epithelial glycans to the glycan receptors present on the sperm apical surface [2,7,25]. Porcine sperm exhibited specific binding affinity to two glycan motifs when incubated with an array of 377 glycans. Each glycan that bound sperm contained either biantennary structures with a mannose core and 6-sialylated lactosamine at one or more termini (bi-SiaLN) or the Lewis X trisaccharide (Le^X^) [7,25], both of which are highly abundant in porcine isthmic epithelium. Interestingly, bi-SiaLN is found throughout the isthmus and ampulla but Le^X^ is limited to the isthmus [7,25]. Further studies showed that Le^X^ and bi-SiaLN motifs of oviduct glycans had affinity for ADAM5 (A Disintegrin and Metalloproteinase Domain 5) and lactadherin (MFGE8, SP47, or SED1) proteins located on porcine sperm head [26]. Additional studies on the function of ADAM5 and lactadherin are necessary to provide insight into whether these are authentic glycan receptors.

The two players in sperm storage are the oviduct cells, which maintain the sperm receptor molecules, and sperm, which possess the cognate receptor(s) that bind to the oviduct. Among the several proposed mechanisms for sperm release from the reservoir are changes in oviduct fluid composition [27], changes in oviduct fluid volume [28], peristaltic contractions of the oviduct [27], alterations in oviduct epithelial cell gene transcription [29], induction of sperm hyperactivation [30], or degradation of sperm glycan receptors [31] facilitating disassociation and release of sperm from reservoir [30]. It may be possible that sperm release from the oviduct reservoir is regulated by a combination of these and other unknown mechanisms.

Released sperm are capacitated and hyperactivated [32,33], although it is not clear if capacitation causes sperm release or sperm release completes capacitation. Additionally, a loss of sperm binding sites on the oviduct epithelium could play a role in sperm release [11]. The signals that induce sperm release and the mechanism behind the detachment of sperm from the oviduct, e.g., loss, alteration, or migration of oviduct binding ligand, are not clear.

One signal that causes sperm release from the oviduct is progesterone [30,34]. Progesterone may act on sperm in the reservoir inducing Ca^2+^ influx and hyperactivated motility [35]. The additional force generated by hyperactivated motility may cause dissociation of sperm from oviduct epithelial cells [36]. In this paradigm, the source of progesterone is uncertain, and it may be secreted from multiple cell types.

In this review, we discuss the sources of progesterone in the oviduct, the concentrations of progesterone in the oviduct in varied conditions across species, and the mechanisms of progesterone-induced sperm release. We include information from oviduct biology and both female and male gamete function to illuminate the sequence of events that culminate in mature sperm meeting the oocyte(s). Key findings are summarized in Table 1.

## 2. Progesterone Facilitates Sperm Release from the Oviduct

Progesterone is a steroid hormone that is produced by adrenal glands and the theca and luteinized granulosa cells (the corpus luteum) in the ovary in the absence of pregnancy and in the placenta and luteinized granulosa cells of the corpus luteum during pregnancy [46]. In the ovary, ovulated follicles are converted into corpora lutea that produce a large amount of progesterone, resulting in an elevated concentration of progesterone in the post-ovulatory oviduct. In pigs, serum concentrations of progesterone begin increasing 6 hr following ovulation and are about 2 ng/mL [37,47]. Progesterone concentrations in the oviduct are expected to be higher than in the serum [48].

In addition to the functions of progesterone during diestrus and pregnancy, there are reports of progesterone function during fertilization. Progesterone has been reported to promote capacitation and induce the acrosome reaction in capacitated sperm from several species [49,50,51,52]. From a fertility point of view, the effects of progesterone may facilitate oocyte fertilization if the acrosome reaction occurs near oocytes. In contrast, premature induction of the acrosome reaction and loss of the acrosomal contents well before sperm reach the oocyte may negatively affect fertilization [53].

The role of progesterone in sperm release from the oviduct epithelium was examined in a classic study. Hunter injected microdroplets of a progesterone solution (10 mg/mL in oil) into the pig oviductal wall surrounding the sperm reservoir 8–12 h before ovulation and 2 h after intracervical insemination. This increased the number of released sperm and induced 3-fold higher polyspermic fertilization [54]. A follow-up study introduced microdroplets of progesterone-rich follicular fluid, containing 600–660 ng/mL progesterone, directly into the pig sperm reservoir and recovered eggs from the ovarian end of the oviduct by flushing the oviduct with a needle inserted from the ampulla. The recovered eggs were analyzed for their numbers of surface-associated sperm along with sperm heads embedded in the zona pellucida. The results show a significantly higher incidence of polyspermy [39]. In addition to demonstrating that the sperm reservoir functions to reduce polyspermy, these results show that sperm were released from the reservoir in higher numbers in response to elevated progesterone in vivo. This was recently reproduced in vitro using porcine sperm bound to oviductal cell aggregates. Progesterone concentrations of 25 ng/mL or 252 ng/mL resulted in sperm release from oviduct cell aggregates within 30 min [34]. Further, evidence from both porcine and bovine studies of oviduct epithelial cell culture, explants, and in vivo studies have provided strong support for the conclusion that progesterone activates sperm release. Progesterone concentrations ranging from 10 ng/mL to 1000 ng/mL induce the release of 32–47% of sperm from bovine oviduct epithelial cells in monolayer culture (BOEC) [40,55]. Romero-Aguirregomezcorta et al. also show that a 3.14 ng/mL concentration of progesterone significantly increases bovine sperm release from oviduct cell explants [41].

The progesterone-induced release of sperm from the reservoir is also observed in other species such as birds [42,56]. Female Japanese quail were injected with progesterone 1 h after mating to bring the estimated serum progesterone concentrations to 1, 10, or 100 ng/mL [42]. The analysis of sperm storage tubules showed that the percentage of tubules with sperm was significantly reduced when birds had progesterone concentrations ≥ 10 ng/mL [42]. Thus, it can be postulated that progesterone is a major factor that induces sperm release from the reservoir in both mammals and birds.

## 3. Sources of Progesterone in the Mammalian Oviduct

The main source of progesterone in the female reproductive system is the corpus luteum. A high concentration of progesterone in oviduct blood, collected from arterioles entering the isthmus, was associated with the number of preovulatory follicles or developing corpus lutea in the nearby ovary, suggesting that ovarian follicles and corpus lutea are the sources of progesterone in oviduct fluid [57]. A higher concentration of progesterone could reach the neighboring oviduct through a counter-current circulation (Figure 1), which has been described [38,58]. Progesterone produced by the ovary could be carried into the ovarian vein and then to the uterine artery and oviduct. The enhanced progesterone concentration could have a major influence on the oviduct lumen. Indeed, in pigs, the concentration of progesterone in arterioles supplying the oviduct is higher than the progesterone concentration in the systemic circulation [38,48].

Progesterone may be synthesized by the oviduct epithelium in several species such as horses and rabbits. In the equine oviduct, enzymes of the steroid biosynthesis pathway such as StAR, cytochrome P450scc, aromatase, and 3β-hydroxysteroid dehydrogenase, identified by antibodies, were present in the epithelium, suggesting local production of steroids [58]. Synthesis of progesterone by the oviduct epithelium and associated smooth muscle cells is also seen in the rabbit [60].

Another potential source of progesterone is the cumulus cells that surround the ovulated oocyte, together termed the cumulus–oocyte complex (COC). These cumulus cells remain with the oocyte during the transport of the oocyte in the oviduct [61,62] and continuously make and secrete progesterone [63,64,65,66,67]. It has been shown that COC’s increase sperm release from the reservoir [43,68]. However, COC’s are in the ampulla and the sperm reservoir is primarily in the isthmus. The distance between the COC and the isthmus dilutes progesterone produced by COC’s. Thus, it seems more likely that COC’s induce release of sperm bound to the nearby ampullary epithelium than the more distant isthmic epithelium.

One attractive hypothesis is that there are at least two sources of progesterone that affect sperm release. First, a basal concentration of progesterone in the oviduct is produced primarily by the ovary and carried to the entire oviduct by counter-current circulation. There, it triggers the release of sperm from some or all of the oviduct epithelium. However, superimposed on the release induced by ovarian progesterone could be progesterone produced by the COC’s that could act in a paracrine manner to trigger release of the fewer sperm bound to the ampulla that are near the COC’s. This hypothesis needs further testing.

## 4. Concentration of Progesterone in the Mammalian Oviduct

Regardless of the source, for progesterone to activate sperm release, it must reach concentrations sufficient to alter sperm function. The oviductal fluid progesterone concentration changes with the stage of the estrous cycle. The elevated level of postovulatory progesterone in the oviduct may be influenced by follicular fluid released into the oviduct, counter-current circulation of progesterone from the ovary, local synthesis of progesterone in the oviduct, and/or the contribution of progesterone from cumulus cells in a paracrine manner.

In cows, the oviduct ipsilateral to the corpus luteum-bearing ovary during the luteal phase and early pregnancy had higher progesterone concentrations, 235.18 ng/g and 362.12 ng/g, respectively, than the contralateral side [69]. However, these differences were less apparent during the follicular phase or immediately post-ovulation. This suggests that much of the additional progesterone in the oviduct produced by the mature corpus luteum is retained locally. The authors also noted a significant elevation of progesterone in the post-ovulatory ipsilateral oviduct. Similarly, in mares, the concentration of progesterone at the postovulatory stage, identified by the presence of corpus hemorrhagicum or recent corpus luteum without uterine edema, is nearly 40-fold higher in the ipsilateral oviduct compared to the contralateral oviduct [58]. This was not observed in oviducts from preovulatory animals. Progesterone concentration was lower during the preovulatory stage compared to the postovulatory stage in both ipsilateral and contralateral oviducts [58]. This study also measured the concentration of progesterone in oviduct fluid. During the early postovulatory stage, progesterone concentration of 103.087 ng/mL was observed in the ipsilateral oviduct fluid, which was 6.5-fold higher than on the contralateral side and 31.2-fold higher than during the preovulatory stage [58], consistent with other observations that oviduct fluid progesterone concentration is elevated following ovulation. These authors proposed that the speedy increase in oviduct progesterone was due to oviduct capture of follicular fluid high in progesterone. Lamy et al. also supported the above results, showing that the oviduct luminal fluid progesterone concentrations increase from 6 ng/mL before ovulation to 57 ng/mL 1–5 days after ovulation in the ipsilateral bovine oviduct [70]. There are no reports of progesterone concentration in luminal fluid from the human Fallopian tube. However, there are studies showing that human peritoneal [71] and follicular fluid [72] progesterone concentrations are higher than the serum progesterone, and it has been suggested that increased peritoneal and follicular fluid progesterone concentrations resemble the post-ovulatory progesterone concentration in the Fallopian tube [73]. The major conclusion to be drawn from this evidence is that the progesterone concentration in the post-ovulatory oviduct is elevated soon after ovulation in both oviducts in litter-bearing animals and on the ipsilateral side in mono-ovulatory animals.

## 5. Concentration of Progesterone That Induces Sperm Release

Sperm release from oviduct epithelial cells by progesterone occurs at a wide variety of hormone concentrations, ranging from 10 to 1000 ng/mL [40]. This report measured release from monolayers of BOEC; there is some concern that oviduct cells may lose some epithelial characteristics in the longer-term cultures used to produce monolayers. To study the role of progesterone in releasing porcine sperm from the oviduct epithelial cells that were collected and tested within a few hours, Machado et al. added porcine sperm to oviduct epithelial cell aggregates and allowed them to bind to the cells for 45 min. These complexes were then treated with either 25 ng/mL or 252 ng/mL of progesterone at 39 °C for 30 min. Progesterone treatment significantly decreased the number of sperm bound to the oviduct epithelial cells at both concentrations compared with controls [34]. Thus, it appears that sperm release from the oviduct epithelium may be triggered by a range of progesterone concentrations. It is also interesting that, despite testing wide ranges of progesterone concentration, nearly all experimental results show incomplete release of sperm. This implies that some sperm are released by other signals. Heterogeneity of sperm, more generally, has been reported [74]. Heterogeneity of sperm release may be a way to avoid completely uniform sperm release, which would be undesirable if that release is much before or well after ovulation.

## 6. Mechanism of Progesterone-Induced Sperm Release from the Oviduct Epithelium

### 6.1. Progesterone-Induced Hyperactivated Motility

The actions of progesterone on sperm to induce release may be unique. The major mode of action of progesterone in somatic cells is via transcriptional regulation through nuclear progesterone receptors that act as transcription factors. Because sperm are transcriptionally silent, regulating gene transcription of sperm by progesterone is not a possibility. It is thought that progesterone regulates biological activities in sperm in a non-genomic fashion via membrane-initiated signaling events [75,76]. Decades ago, it was discovered that progesterone stimulates a rapid Ca^2+^ influx in the human sperm and that a membrane-impermeant version also induced Ca^2+^ influx [77]. More recent work has demonstrated that progesterone induces human sperm Ca^2+^ influx by binding to a serine hydrolase in the sperm plasma membrane, ABHD2, that, in turn, depletes sperm membrane 2-arachidonyl glycerol, releasing a sperm-specific calcium channel called CatSper, from inhibition [44,78]. Thus, ABHD2 acts as a nongenomic progesterone receptor in human sperm [79] (See Figure 2 for model).

Progesterone induces hyperactivated motility of mammalian sperm [49,68]. The major characteristic of hyperactivated motility is increased flagellar bending amplitude, which causes increased asymmetrical beat pattern and swimming trajectories that are either helical or circular [84]. Hyperactivation increases the force that the sperm tail generates [85,86] and, perhaps by doing so, overcomes the overall binding avidity of sperm for oviduct epithelial cells, inducing sperm release from the oviduct reservoir [87].

CatSper channels are essential for the development of hyperactivated motility in human, rhesus macaque, and mouse sperm [68,88,89,90,91,92]. Interestingly, progesterone opens CatSper channels in human sperm, but not in mouse sperm [93,94]. That CatSper is activated differently in sperm from certain species is notable. It suggests that progesterone may be less important in releasing mouse sperm from the oviductal reservoir than human sperm or perhaps macaque and porcine sperm. Although patch-clamp studies have not been carried out, the observation that inhibitors of ABHD2 and CatSper affect the response to progesterone by porcine sperm [34] suggests that progesterone acts on CatSper in sperm from species in addition to humans and monkeys.

Several signaling events, presumably downstream of the progesterone-induced Ca^2+^ influx, have been described (Figure 2). Progesterone-induced elevation of Ca^2+^ activates soluble adenylyl cyclase (sAC) which, in turn, increases the activity of Protein Kinase A (PKA), increasing sperm protein serine/threonine phosphorylation and leading to sperm capacitation and hyperactivated motility [80]. Progesterone also stimulates the Protein Kinase C (PKC), MAPK [75,76,95,96,97,98], and Janus Kinase pathways [99] in sperm from several species including humans. In addition, inhibition of tyrosine kinases and PKA alters progesterone-induced sperm chemotaxis in human sperm [100]. This suggests that progesterone-induced sperm hyperactivated motility is associated with sperm protein kinases that become activated following Ca^2+^ influx. Finally, there is evidence of a function for calmodulin (CaM)-dependent kinases in the sperm response to increased Ca^2+^. A study using demembranated bovine sperm showed that hyperactivated motility can be restored by increasing free Ca ^2+^ in reactivation medium to 1 µM. However, when the reactivation medium is supplemented with anti-CaM IgG, the sperm hyperactivated motility in 1 µM Ca^2+^ is reduced significantly. Further, when the reactivation medium is supplemented with CaM kinase II-inhibiting peptides, hyperactivated motility is significantly reduced [101]. These results suggest that sperm hyperactivated motility is regulated by the Ca^2+^/CaM kinase II pathway [101,102].

The entire process of capacitation includes hyperactivation as well as other plasma membrane and metabolic changes that mammalian sperm go through in the female reproductive tract to become competent to fertilize oocytes [103,104,105,106]. Capacitation is associated with sperm Ca^2+^ influx [106,107], protein tyrosine phosphorylation [108], cholesterol efflux [104,108], and increased sperm membrane fluidity [103,109]. Even though hyperactivation is a component of capacitation, certain conditions allow for separation of the two processes [110,111]. Studies that separate the two processes have found that hyperactivation is not always sufficient to induce sperm release from the reservoir; other sperm capacitation-related changes may contribute to progesterone-induced sperm release [27,84,112]. Before capacitation, sperm are capable of binding to the oviductal epithelium, but only a few sperm bind to the oviduct epithelium following capacitation [32]. This is consistent with the loss of the ability of capacitated sperm to bind soluble fluorescent 3-O sulfated Le^X^ and bi-SiaLN [7,25]. Because hyperactivated motility would not affect the binding of soluble glycans, capacitated sperm have lost their affinity for these two glycan motifs, which would further induce sperm release. The loss of biding to soluble oviduct glycans results suggest that progesterone induces a loss in glycan affinity, independent of its effects on sperm hyperactivation.

### 6.2. Progesterone-induced Activation of the PI3K/AKT Pathway

Progesterone may have effects on sperm other than Ca^2+^ influx. For example, it was proposed that progesterone activation of PI3K and phosphorylation of AKT is involved in the alteration of sperm motility including hyperactivated motility [113] induced during capacitation. Activation of PI3K may be downstream of CatSper channels [113]. According to Sagare-Patil et al., under in vitro conditions, relatively high (1.57 µg/mL and 3.14 µg/mL) concentrations of progesterone increased human sperm AKT phosphorylation following 10 min of incubation [45]. Pujianto et al., showed that 750 ng/mL of progesterone increased human sperm AKT phosphorylation [114]. Further, the inhibition of PI3K is associated with a decrease in sperm motility and impaired hyperactivation of hamster sperm [115] and boar sperm [116]. How opening CatSper channels may activate the PI3K/AKT pathway and the steps after AKT phosphorylation are still unknown [113]. The potential role of the PI3K/AKT pathway in sperm release is uncertain.

### 6.3. Progesterone and Sperm Membrane Modifications

Binder of sperm proteins, BSP-1, BSP-3, and BSP-5, from the seminal plasma, that bind/coat bovine sperm surface at the time of ejaculation are involved in sperm binding to oviduct epithelial cells [117]. Removing BSPs from the sperm surface by components in the oviduct fluid or degradation of BSPs promotes cholesterol efflux, leading to increased membrane fluidity, an initiating factor in sperm capacitation [118]. Progesterone-induced sperm release from BOEC also results in sperm with a lower abundance of BSP-3 and BSP-5 and increased membrane fluidity [55]. This suggests that progesterone can induce a loss of peripheral membrane proteins causing sperm release from oviduct cells. The mechanism by which this is accomplished is unclear.

### 6.4. Potential Effects of Progesterone on Sperm Glycan Receptors

Following capacitation, porcine sperm lose their ability to bind both glycan motifs that act as sperm receptors on oviduct cells [7,25]. This may be a result of glycan receptor degradation, shedding, sperm membrane modifications, or a combination of these events [30]. Receptor degradation may be accomplished by sperm proteasomes. Miles et al. created a transgenic pig that carries GFP-tagged 20S proteasomal core subunit α type 1 (PSMA1-GFP) and showed that this fusion protein is incorporated into sperm proteasomes. Precipitation experiments were performed to identify sperm proteins that interact with PSMA1-GFP, and sperm candidate glycan receptors lactadherin and ADAM5 were identified among proteins that were precipitated with PSMA1. Their work suggests an association of glycan receptors lactadherin and ADAM5 with the ubiquitin–proteasome system that might degrade these candidate receptors and promote sperm release. Supporting this hypothesis, Sharif et al. showed that the inhibition of proteasomal degradation results in a reduction in sperm release from oviduct glycans [30].

Further, human sperm proteasomal activity increases during capacitation, and the increase requires the SACY/cAMP/PKA pathway [119]. Progesterone also activates the SACY/cAMP/PKA pathway in human sperm [98], which may further increase the action of the ubiquitin–proteasome system, perhaps leading to degradation of the glycan receptor proteins (Figure 2). Thus, in addition to inducing hyperactivation, progesterone may also promote sperm release by activating sperm glycan receptor degradation [30].

## 7. Conclusions

The action of progesterone on sperm Ca^2+^ regulation has been clarified over the last 20 years, particularly in human sperm. Through activation of a non-genomic receptor, progesterone can open CatSper channels, sperm-specific Ca^2+^ channels, and increase the percentage of sperm displaying the hyperactivated form of motility. There is evidence that sperm from some other species including, macaque and porcine, follow the paradigm of human sperm. Thus, progesterone can induce hyperactivation, which may drive sperm release by several mechanisms including increasing the force necessary to tether sperm. Progesterone action as a releasing signal may also be conserved in birds [42]. However, there is also evidence for additional mechanisms that progesterone may activate to promote sperm release, including sperm protein shedding or hydrolysis. There may be a combination of signals involved in the induction of sperm release.

An attractive but relatively untested hypothesis is that signals such as progesterone promote a basal amount of sperm release from the reservoir during the peri-ovulatory period. This progesterone may be released from the ovary via a counter-current circulation. A second source of progesterone superimposed on the basal release and more closely timed to ovulation may facilitate further release of sperm. Together, the amount of progesterone is sufficient to release a high number of sperm from the reservoir for fertilization.

Future studies should focus on clarifying if progesterone is involved in overcoming the connection between sperm glycan receptors and oviduct glycans and/or if progesterone is involved in partial or complete proteolysis of sperm glycan receptors. It is also important to identify the signaling pathways that progesterone binding activates to promote sperm release from oviduct glycans. Advances in these topics could lead to higher fertility by better synchronizing sperm release with the presence of the cumulus–oocyte complex.

## Figures and Tables

**Figure 1 cells-11-01622-f001:**
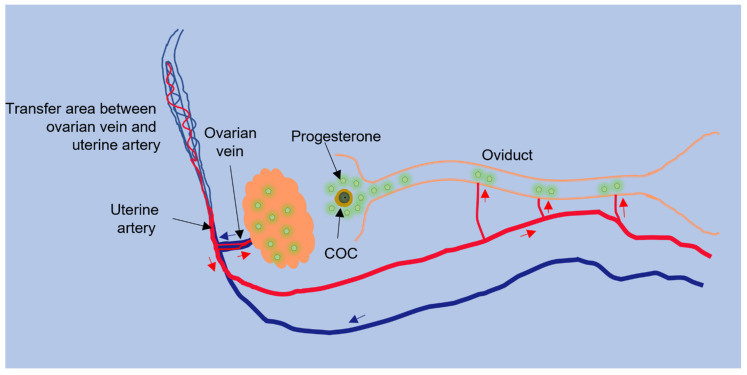
Counter-current transfer of progesterone. The oviductal sperm reservoir may be acted upon by progesterone from the ovary via counter-current circulation. Progesterone produced by the ovary can move into the ovarian vein and then the uterine artery that supplies blood to the oviduct, resulting in progesterone produced in the ovary undergoing less diffusion before reaching the oviduct (modified from [59]). The arrows indicate blood flow in the arterial (red) or venous (blue) system.

**Figure 2 cells-11-01622-f002:**
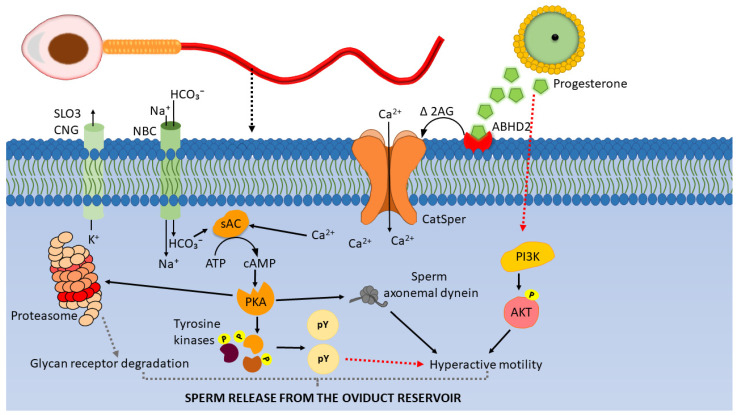
Working model of signaling events leading to sperm release from the oviductal reservoir. Progesterone binding to ABHD2 receptors on the sperm membrane depletes 2-arachidonoyl glycerol (2AG) and opens the CatSper channels permitting Ca^2+^ entry. Ca^2+^ and HCO_3_^-^ activate sAC which produces cAMP from ATP. cAMP activates Protein Kinase A (PKA). PKA (i) phosphorylates and activates sperm tyrosine kinases [80,81]. Tyrosine kinases phosphorylate the tyrosine residues of sperm proteins which leads to hyperactivated motility, (ii) associate with axonemal dynein, phosphorylate them and regulate hyperactivated motility, (iii) phosphorylate sperm 26S proteasome subunits and activate proteasome activity, which may result in degradation of sperm glycan receptors. The combination of hyperactivated motility and glycan receptor degradation may facilitate sperm detachment from the oviduct sperm reservoir (modified from [82,83]).

**Table 1 cells-11-01622-t001:** Summary of key findings.

Key Findings	Species	Reference
Progesterone induces hyperactivated motility of mammalian sperm	Human	[35]
Sperm capacitation and hyperactivation result in sperm detachment from the oviduct epithelium	Hamster	[32]
Hyperactivated motility causes detachment of sperm from oviduct epithelial cells	Human	[36]
Blood serum concentrations of progesterone from utero tubal junction–isthmic region increases following ovulation	Pig	[37]
Higher concentration of progesterone could reach the neighboring oviduct through counter-current circulation	Pig	[38]
Injecting progesterone-rich follicular fluid to sperm reservoir increases the incidence of polyspermic fertilization	Pig	[39]
Progesterone concentrations of 25 ng/mL or 252 ng/mL induce sperm release from oviduct cell aggregates	Pig	[34]
Progesterone concentrations from 10 ng/mL to 1000 ng/mL induce the release of 32–47% of sperm from oviduct epithelial cells in monolayer culture	Bovine	[40]
3.14 ng/mL (10 nM) concentration of progesterone increases sperm release from oviduct cell explants	Bovine	[41]
Progesterone induced release of sperm from the reservoir in birds	Japanese quail	[42]
Cumulus–oocyte complex increases sperm release from the reservoir	Pig	[43]
ABHD2 acts as a nongenomic progesterone receptor in sperm	Human	[44]
High concentration of progesterone triggers activation of multiple kinase pathways in sperm, hyperactivation and acrosome reaction	Human	[45]
Low concentrations of progesterone triggers calcium influx and tyrosine kinase activation	Human	[45]
Inhibition of proteasomal degradation results in reduction in sperm release from oviduct glycans	Pig	[30]

## Data Availability

Not applicable.

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
