# Peer review of "Progesterone-Induced Sperm Release from the Oviduct Sperm Reservoir"

_cells, 2022, doi:10.3390/cells11101622_

Round 1

Reviewer 1 Report

The storage of sperm in oviduct isthmus is important for mammals to increase the fertilization possibility during ovulation. Although many factors may help the release of sperm from this reservoir, the contribution of progesterone is particularly interesting because its concentration is largely increased around ovulation and it induces sperm hyperactivation which enables the sperm to detach from the oviduct epithelial cells. Accordingly, the authors summarized the research progress regarding the sources of progesterone in the mammalian oviduct, the evidences supporting the importance of progesterone for sperm release, together with the underlying mechanisms by which progesterone facilitate sperm release. Overall, this review was well written and covered the most recent advances in this field. I suggest the authors discuss/clarify the following points in the revised version.

  1. As the authors stated in the manuscript, the calcium influx through CatSper induced by progesterone is critical for human sperm hyperactivation, which may be a major player during progesterone-caused sperm release from isthmus. However, it seems that progesterone doesn’t activate mouse CatSper. Does this mean that progesterone is more important for human sperm to detach from the reservoir than mouse sperm? Please add discussion on this concern.
  2. In Figure 2, the authors speculated that besides CatSper HCN is also a resource for calcium influx in mammalian sperm. However, the related evidence is limited. Please add pertinent references to support this claim.

Author Response

    1. As the authors stated in the manuscript, the calcium influx through CatSper induced by progesterone is critical for human sperm hyperactivation, which may be a major player during progesterone-caused sperm release from isthmus. However, it seems that progesterone doesn’t activate mouse CatSper. Does this mean that progesterone is more important for human sperm to detach from the reservoir than mouse sperm? Please add discussion on this concern.

    Yes it may mean that progesterone is less important for mouse sperm release from the oviduct than sperm from humans and perhaps other animals. This point has been added to the manuscript (lines 265-273).

    1. In Figure 2, the authors speculated that besides CatSper HCN is also a resource for calcium influx in mammalian sperm. However, the related evidence is limited. Please add pertinent references to support this claim.

    Thank you.  We have removed HCN from the figure.

Reviewer 2 Report

It is well known that progesterone(P4) can induce the acrosome reaction(AR) in capacitated sperm. It was described here that P4 can induce sperm capacitation, hyperactivated motility and release from the oviduct epithelial cells. Thus, after this release, P4 might induce the AR far from the egg, a process that might abrogate fertilization. I think that the authors should relate to this possible situation.

Author Response

It is well known that progesterone(P4) can induce the acrosome reaction(AR) in capacitated sperm. It was described here that P4 can induce sperm capacitation, hyperactivated motility and release from the oviduct epithelial cells. Thus, after this release, P4 might induce the AR far from the egg, a process that might abrogate fertilization. I think that the authors should relate to this possible situation.

Yes progesterone could induce the acrosome reaction, if the concentrations are adequate. We have added some discussion of this possibility in lines 109-115.

Reviewer 3 Report

In this Manuscript the authors review the potential mechanisms of mammalian sperm release from the sperm reservoir in the female reproductive tract focusing specifically on progesterone-induced effects. This is a very interesting and relevant topic, that deserves wider attention. Some comments:

1- In the Introduction the authors state (l 36) " We propose here that a basal number of sperm..." which seems an odd statement, both for a Review, and for an Introduction.

2- While mostly described it is not always clear in some instances what species the authors are discussing. For example progesterone-induced signalling pathways, hyperactivation and many other instances. This should be clear to a non-specialist reader. Moreover species-specific differences should be discussed, or what data is missing in certain key species, or other aspects related to this crucial issue in reproduction.

3-In this regard it would be useful if the authors systematized the review, by describing the literature more fully at the beginning, what searches were performed in which databases, how many studies were found etc.

4-In addition a Table summarizing  the key findings (what was found in each paper, what species was used, etc.) would be important besides the narrative text, where comparisons and overall views are hard to determine.

5- The discussion of progesterone-dependent signalling in human sperm is interesting, but I found no information of the concentration of progesterone in the human reproductive tract (not necessarily the oviduct). Or on the whole concept of sperm reservoir in humans. Without this information how relevant is this particular data for the specific focus of this review? What data is really important in terms of reservoir release, and not for other aspects of sperm function after it is released? In reading this part of the text it seemed that their focus of the review was not what the authors intended and listed. A review on the specific action of progesterone on sperm (not in terms of reservoir in the female tract) is also a totally distinct possibility for a review, of course.

6- Right at the beginning a more detailed discussion of the whole concept of sperm reservoir and how it is thought to act in several species would be very useful for a non-specialized reader.

Author Response

1- In the Introduction the authors state (l 36) " We propose here that a basal number of sperm..." which seems an odd statement, both for a Review, and for an Introduction.

Initially, we put the statement there to introduce it early and lay the groundwork for more detailed discussion later in the manuscript. But we have now removed this statement.

2- While mostly described it is not always clear in some instances what species the authors are discussing. For example progesterone-induced signalling pathways, hyperactivation and many other instances. This should be clear to a non-specialist reader. Moreover species-specific differences should be discussed, or what data is missing in certain key species, or other aspects related to this crucial issue in reproduction.

We have now included the species in these discussions.

3-In this regard it would be useful if the authors systematized the review, by describing the literature more fully at the beginning, what searches were performed in which databases, how many studies were found etc.

We have added a table that includes key papers, based on literature searches and our knowledge of the field. The Table also includes the species in which the findings were made, in accordance with Comment 2 from Reviewer 3 above.

4-In addition a Table summarizing  the key findings (what was found in each paper, what species was used, etc.) would be important besides the narrative text, where comparisons and overall views are hard to determine.

This is included in Table 1.

5- The discussion of progesterone-dependent signalling in human sperm is interesting, but I found no information of the concentration of progesterone in the human reproductive tract (not necessarily the oviduct). Or on the whole concept of sperm reservoir in humans. Without this information how relevant is this particular data for the specific focus of this review? What data is really important in terms of reservoir release, and not for other aspects of sperm function after it is released? In reading this part of the text it seemed that their focus of the review was not what the authors intended and listed. A review on the specific action of progesterone on sperm (not in terms of reservoir in the female tract) is also a totally distinct possibility for a review, of course.

We have included more discussion of human sperm and the possible formation of a sperm reservoir (lines 52-60). We tried to keep the review focused on the effects of progesterone on sperm release from the reservoir to avoid it becoming too long. But we do introduce other effects of progesterone on sperm, including their ability to promote capacitation and the acrosome reaction (lines 109-115).  Our goal was to prepare a more focused review rather than one that discussed all effects of progesterone on sperm, including those after sperm release.

6- Right at the beginning a more detailed discussion of the whole concept of sperm reservoir and how it is thought to act in several species would be very useful for a non-specialized reader.

We have added this information at the beginning of the Introduction, pages 43-51.

Round 2

Reviewer 3 Report

The authors have addressed my main comments. I have no further issues.